# Antiviral Effects of Adipose Tissue-Derived Mesenchymal Stem Cells Secretome against Feline Calicivirus and Feline Herpesvirus Type 1

**DOI:** 10.3390/v14081687

**Published:** 2022-07-30

**Authors:** Takahiro Teshima, Yuyo Yasumura, Ryohei Suzuki, Hirotaka Matsumoto

**Affiliations:** 1Laboratory of Veterinary Internal Medicine, Department of Veterinary Clinical Medicine, School of Veterinary Medicine, Faculty of Veterinary Science, Nippon Veterinary and Life Science University, 1-7-1 Kyonan-cho, Musashino 180-8602, Tokyo, Japan; d2203@nvlu.ac.jp (Y.Y.); ryoheisuzuki@nvlu.ac.jp (R.S.); matsumoto@nvlu.ac.jp (H.M.); 2Research Center for Animal Life Science, Nippon Veterinary and Life Science University, 1-7-1 Kyonan-cho, Musashino 180-8602, Tokyo, Japan

**Keywords:** cat, feline calicivirus, feline herpesvirus type 1, mesenchymal stem cell, secretome, viral replication

## Abstract

Mesenchymal stem cells (MSCs) have excellent anti-inflammatory and immunomodulatory capabilities and therapeutic effects in some viral diseases. The therapeutic impact of MSCs mainly relies on the paracrine effects of various secreted substances. Feline calicivirus (FCV) and feline herpesvirus type 1 (FHV1) are common and highly prevalent pathogens causing upper respiratory diseases, and FCV is associated with gingivostomatitis in cats. Recently, feline MSC treatment has been reported to improve the clinical symptoms of feline chronic gingivostomatitis, but the antiviral effects of feline MSCs on FCV and FHV1 are not known. In this study, we evaluated the antiviral efficacy of using feline MSC secretome as a conditioned medium on FCV and FHV1 viral replication in Crandell–Reese feline kidney (CRFK) cells, and RNA sequencing was used to analyze how the CRFK cells were altered by the MSC secretomes. The feline MSC secretome did not inhibit FCV or FHV1 viral entry into the CRFK cells but had antiviral effects on the replication of both FCV and FHV1 in a dose-dependent manner.

## 1. Introduction

Mesenchymal stem cells (MSCs) can be isolated from a variety of tissues, such as bone marrow, umbilical cord, and adipose tissue. Adipose tissue-derived MSCs (AD-MSCs) are easy to harvest, have excellent proliferation potential, and are progenitor cells that harbor valuable therapeutic and biological activities [1,2]. Studies have revealed the many abilities of AD-MSCs, such as reducing inflammation, modulating immune responses, and promoting tissue regeneration [3,4,5]. The therapeutic capacities of MSCs mainly rely on the variety of their secretomes [6,7,8]. Recently, several clinical studies involving the use of MSCs and/or their secretomes to treat viral infection in human beings, such as hepatitis B virus, human immunodeficiency virus, and severe acute respiratory syndrome coronavirus 2, have been registered [9].

Feline calicivirus (FCV) is a highly contagious pathogen of cats that is found worldwide and causes upper respiratory tract and oral disease [10]. Feline herpesvirus 1 (FHV1) is one of the most common viruses among cats, and FHV1 infection is associated with respiratory and ocular diseases [11]. Vaccinations against these viruses are widespread throughout the world, but they are not fully effective [12,13]; thus, it is of urgency that we develop effective and safe antiviral drugs for veterinary medicine. Commercial formulation of recombinant type I interferons (IFNs) is sometimes use against FCV. IFNs are involved in antiviral responses and have broad spectrum antiviral activities [14]; however, many viruses have required multiple strategies to escape or inhibit the IFN response [15,16,17]. Moreover, some FCV strains do not promote the activation of the IFNβ promoter, allowing these viruses to evade the IFN response [18]. Antiviral drugs against FHV1 have also been examined for their effects on plaque number and plaque size in vitro, and it was demonstrated that ganciclovir, 9-(2-phosphonylmethoxyethyl)-2,6-diaminopurine (PMEDAP), and cidofovir are the most potent inhibitors of FHV1 replication in Crandell–Reese feline kidney (CRFK) cells [19].

Feline chronic gingivostomatitis (FCGS) is a multifactorial disease and FCV is one of the causes associated with it. FCGS-affected cats with FCV and puma feline foamy virus may poorly respond to treatment [20]. Recently, feline chronic gingivostomatitis (FCGS) was reportedly treated with AD-MSCs [21,22,23]. In the report, feline AD-MSCs administered systemically resulted in favorable clinical, histologic, and systemic responses in over 70% of cats [23]. However, it remains unknown whether feline AD-MSCs have antiviral effects on FCV and FHV1. In this study, the antiviral effects of secretome from feline AD-MSCs on FCV and FHV1 replication were investigated by measuring changes in viral mRNA (vmRNA) levels and virus yields in CRFK cells. Then, gene expression was performed using RNA sequencing to detect any possible alterations related to the inhibition of these viruses between the mock CRFK cells and the CRFK cells sensitized to feline AD-MSC secretome.

## 2. Materials and Methods

### 2.1. Isolation and Expansion of Feline AD-MSCs

Adipose tissue was aseptically collected from falciform ligament fat of three healthy anaesthetized client-owned Japanese bobtail short-hair cats (female, mean body weight of 3.3 kg, mean age of 9 months) when they were scheduled to be spayed. The tissue was washed extensively in a phosphate buffer solution (PBS), minced, and digested with collagenase type I (Sigma-Aldrich) at 37 °C for 45 min with shaking. After washing with PBS and centrifuging, the pellets containing the stromal vascular fraction were resuspended, filtered through a 100-μm nylon mesh, and incubated overnight in Dulbecco’s modified Eagle’s medium (DMEM) supplemented with 10% fetal bovine serum (FBS; Capricorn Scientific) and a 1% antibiotic–antimycotic solution (Thermo Fisher Scientific) in a humidified atmosphere with 5% CO_2_ at 37 °C. Unattached cells were removed by changing the medium, and the attached cells were washed twice with PBS. Thereafter, the medium was replaced every 3–4 days. At 80–90% confluence, the cells were detached with trypsin-EDTA solution (Sigma-Aldrich) and passaged repeatedly.

### 2.2. Characterization of Feline AD-MSCs

The expression of several markers, such as CD14-FITC, CD29-PE, CD34-FITC, CD44-PE, CD45-PE, CD90-PE, CD105-PE, and MHC-II-FITC on passage 2 feline AD-MSCs was determined by flow cytometry using a CytoFLEX (BECKMAN COULTER). The capacity of the cells to differentiate into osteogenic, chondrogenic, and adipogenic lineages was evaluated.

### 2.3. Production of fADSC-CM

In this study, feline AD-MSC secretome was collected as conditioned medium (CM). To prepare the feline AD-MSC CM (fADSC-CM), passage 2 feline AD-MSCs were seeded (3.0 × 10^4^ cells/cm^2^) individually into DMEM supplemented with 10% FBS and 1% antibiotic–antimycotic solution and incubated overnight. Adherent cells were washed and further incubated in FBS-free DMEM for 36 h, then the medium was centrifuged at 300× *g* for 5 min and 1200× *g* for 20 min at 4 °C to remove cells and debris. The naïve CM was concentrated using molecular weight cut-off (MWCO) ultrafiltration with membranes of 3 kDa, 30 kDa, and 100 kDa. The total protein concentrations of naïve and MWCO-concentrated supernatants were measured with a BCA assay kit, then stored at −80 °C until further assay.

### 2.4. Cells and Viruses

The CRFK cells were obtained from the American Type Culture Collection. The cells were cultured in DMEM containing 10% FBS, 100 U/mL penicillin, and 100 µg/mL streptomycin (complete medium). FCV strain F4 and FHV1 strain C7301 were used in this study. The viral titers of FCV and FHV1 were determined by median tissue culture infectious dose (TCID50) assay.

### 2.5. Cytotoxicity Assay for fADSC-CM

Prior to the assay, the CRFK cells (2.0 × 10^4^ cells/well, 100 µL/well) were seeded into flat-bottom 96-well plates and incubated overnight with 5% CO_2_ at 37 °C. The cell monolayers were then treated with a series of protein concentrations (6.25, 12.5, 25, 50, 100, 150, 200 µg/mL) of fADSC-CM for 24 h and 72 h. Cytotoxicity assays were performed using a Cell Counting Kit-8 according (Dojindo) to the manufacturer’s instructions.

### 2.6. Effect of fADSC-CM on Viral Replication

To evaluate whether fADSC-CM would affect the replication of FCV and FHV1 in CRFK cells, cells (2.0 × 10^4^ cells/well) were incubated overnight in 24-well plates at 37 °C and 5% CO_2_ in the complete medium. The complete medium was removed, and the monolayer was infected with FCV and FHV1 at a multiplicity of infection (MOI) of 0.1 at 37 °C for 1 h in DMEM. After removing DMEM containing viruses and washing three times with PBS, the infected cells were incubated with fADSC-CM (50 and 200 µg/mL) for 24 h. After incubation, the total RNA was extracted from the cells, and the levels of vmRNA were determined by real-time reverse-transcription (RT) PCR.

### 2.7. Effect of fADSC-CM on Viral Entry

To evaluate whether fADSC-CM would affect the entry of FCV and FHV1 into CRFK cells, cells were incubated overnight in 24-well plates at 37 °C in 5% CO_2_ in the complete medium. The complete medium was removed, and the monolayer was infected at an MOI of 0.1 with FCV and FHV1 solutions containing different concentrations of fADSC-CM (50 µg/mL and 200 µg/mL) at 37 °C for 1 h. After removing the virus solutions and washing three times with PBS, the cells were incubated in the complete medium for an additional 1 h. After incubation, the total RNA was extracted from the cells to determine the levels of vmRNA by real-time RT-PCR.

### 2.8. Effects of Different Amounts of fADSC-CM on Viral Replication

To determine whether the antiviral effect of fADSC-CM on the CRFK cells was dose-dependent, different concentrations of naïve fADSC-CM (50 µg/mL, 100 µg/mL, 150 µg/mL, and 200 µg/mL) were administrated after FCV and FHV1 infection. After incubation for 24 h at 37 °C in 5% CO_2_, the total RNA was extracted from the cells, and the levels of vmRNA were determined by real-time RT-PCR.

### 2.9. Effect of fADSC-CM on Different Phases of CRFK Cells under Infection Conditions

To compare the antiviral activity of fADSC-CM on the CRFK cells challenged with infection, the CRFK cells were exposed to naïve fADSC-CM before infection, at the time of infection, and/or after infection and incubated in 24-well plates or 12 h at 37 °C in the complete medium. For the before the infection group, the monolayer was incubated with 200 µg/mL of naïve fADSC-CM at 37 °C for 12 h, then infected with FCV and FHV1 at an MOI of 0.1 for 1 h. After infection, the CRFK cells were washed three times with PBS and incubated in 2% FCS DMEM for 24 h. For the group representing the duration of the infection, the monolayer was incubated in the complete medium for 12 h, then exposed to FCV and FHV1 solutions at an MOI of 0.1 that included 200 µg/mL of fADSC-CM for 1 h. For the after the viral infection group, the CRFK cells were washed three times with PBS, then incubated in 2% FCS DMEM for 24 h; then the monolayer was incubated with the complete medium for 12 h and the cells infected with FCV and FHV1 at an MOI of 0.1 for 1 h. After infection, the CRFK cells were washed three times in PBS and incubated in 200 µg/mL of naïve fADSC-CM for 24 h. After incubation for 24 h, the total RNA was extracted from the cells, and the levels of vmRNA were determined by real-time RT-PCR. After 24 h incubation of all the groups, cell supernatants were collected for using the plaque assay.

### 2.10. Plaque Assay

When the CRFK cells in the 10 cm-dish formed a monolayer, the cell supernatants were diluted serially 10-fold and added to the cells after removing the complete medium and washing with PBS. After incubating at 37 °C for 1 h, 2× DMEM and 3% carboxymethyl-cellulose were mixed at a 1:1 ratio and added to the plates, followed by culture at 37 °C for 36 h [24]. The cells were fixed with 4% formaldehyde and stained with 1% crystal violet to calculate the number of plaques.

### 2.11. Real-Time RT-PCR

The total RNA from FCV- and FHV1-infected CRFK cells was isolated using NucleoSpin RNA (Takara) according to the manufacturer’s instructions. cDNA was synthesized from 0.5 µg of the total RNA using random primers and the GoScript Reverse Transcriptase system (Promega), according to the manufacturer’s instructions. Real-time RT-PCR analyses were performed using GoTaq Probe qPCR Master Mix (Promega) to determine the mRNA levels of FCV strain F4 and the glycoprotein C gene for FHV1 strain F7301. mRNA levels of the housekeeping gene β-glucuronidase (*GUSB*) were used for normalization. The primers and probe sequences are shown in Table 1 [25,26,27]. Amplification conditions were 95 °C for 2 min, followed by 40 cycles of 95 °C for 15 s and 60 °C for 60 s. After 40 cycles, a dissociation curve was generated to verify the specificity of each primer. All reactions were performed in duplicate. Expression levels of target genes were normalized to the level of GUSB and quantified by the ΔΔCt method.

### 2.12. RNA Sequencing

We performed RNA-seq of the total RNA samples isolated from the CRFK cells after 12 h of incubation in the complete medium (control, n = 3) and with 200 µg/mL of naïve fADSC-CM (n = 3). cDNA library construction was carried out with 1 µg of the total RNA using NEBNext Ultra II RNA Library Prep Kit for Illumina, according to the manufacturer’s instructions, followed by paired-end sequencing (2 × 150 bp) using the Novaseq6000. For each library, an average of 16–20 million read pairs were generated. Quality control checks of the sequencing raw data were conducted with FastQC, while adapter-trimming was performed with Trim Galore. The relative expression of transcripts was quantified for each sample using featureCount. Fastq files were mapped to the Felis_Catus_9.0 reference genome using STAR software (ver2.7). Differentially expressed genes (upregulated or downregulated genes) were determined using edgeR (ver. 3.22.3) based on an adjusted *p*-value of <0.05 and fold change of >2 or <0.5. Biological function Gene Ontology (GO) analysis was performed by g:Profiler, and gene set enrichment analysis (GSEA) was performed by RaNA-seq.

### 2.13. Statistical Analysis

All experiments were performed in triplicate and repeated in three independent experiments except for RNA-seq, and all data are presented as the mean ± standard deviation. Differences among multiple groups were assessed by one-way analysis of variance, and differences were compared using the Tukey–Kramer post hoc test. *p* < 0.05 was considered statistically significant. Statistical analyses were performed using Excel 2019 with add-in software Statcel 3.

## 3. Results

### 3.1. Characterization of Feline AD-MSCs

Feline AD-MSCs were successfully cultured and expanded. The majority of the cells expressed the established MSC markers CD29 (97.9 ± 0.9%), CD44 (99.8 ± 0.1%), CD90 (99.0 ± 1.1%), and CD105 (97.9 ± 1.1%) and very few expressed CD14 (0.27 ± 0.23%), CD34 (0.34 ± 0.30%), CD45 (0.44 ± 0.08%), or MHC-II (0.22 ± 0.23%). The AD-MSCs exhibited multilineage plasticity as demonstrated by their potential for adipogenic, osteogenic, and chondrogenic differentiation.

### 3.2. Cytotoxicity Assay of CRFK Cells in fADSC-CM

The results of the cytotoxicity assay in the CRFK cells sensitized with fADSC-CM at a concentration of 6.25–200 µg/mL for 24 h or 72 h showed a relative cell viability that was almost greater than that of the mock CRFK cells (Figure 1). Therefore, a concentration of 200 µg/mL fADSC-CM was used as the maximum concentration for the antiviral experiments.

### 3.3. fADSC-CM Concentrated by MWCO Ultrafiltration Inhibits Virus Replication

The viral replication of both FCV and FHV1 were suppressed when cells were incubated in 200 µg/mL of naïve and 3 kDa, 30 kDa, and 100 kDa MWCO-treated fADSC-CM (Figure 2). The relative FCV vmRNA levels in the cells treated with 200 µg/mL of naïve and 3 kDa, 30 kDa, and 100 kDa fADSC-CM were 42.1%, 48.3%, 58.4%, and 61.4%, respectively, compared with those in mock CRFK cells. The relative FHV1 vmRNA levels in the cells treated with 200 µg/mL of naïve and 3 kDa, 30 kDa, and 100 kDa fADSC-CM were 50.9%, 57.4%, 64.9%, and 70.1%, respectively, compared with those in the mock CRFK cells. However, when the concentration of fADSC-CM was decreased to 50 µg/mL, only naïve and 3 kDa fADSC-CM significantly suppressed both FCV and FHV1 replication.

### 3.4. fADSC-CM Does Not Prevent FCV and FHV1 Entry

The vmRNA levels after both FCV and FHV1 infection showed that cell treatment in fADSC-CM at concentrations of 50 µg/mL or 200 µg/mL did not prevent viral entry (Figure 3). With 200 µg/mL of the MWCO 100 kDa concentration of fADSC-CM, the highest vmRNA levels for both FCV (207%) and FHV1 (171%) compared with the mock CRFK cells were seen.

### 3.5. Inhibition of Viral Replication Depends on Amount of Naïve fADSC-CM

The evaluation of the effects of fADSC-CM concentrated by MWCO ultrafiltration showed that naïve fADSC-CM inhibited the viral replication of both FCV and FHV1 the most. Therefore, we investigated whether the antiviral effects of fADSC-CM are associated with its concentration (in the range 50–200 µg/mL). The vmRNA levels of both FCV and FHV1 showed that the highest concentration of naïve fADSC-CM inhibited the most viral replication (Figure 4). The FCV mRNA levels in the CRFK cells sensitized with 50, 100, 150, and 200 µg/mL of naïve fADSC-CM were 64.9%, 56.7%, 45.1%, and 42.1%, respectively, compared with the levels in mock CRFK cells. The FHV1 mRNA levels in the CRFK cells treated with 50, 100, 150, and 200 µg/mL of naïve fADSC-CM were 62.3%, 60.3%, 50.4%, and 48.8%, respectively, compared with those in mock CRFK cells.

### 3.6. Naïve fAD-MSC-CM Treatment Inhibits Viral Replication of Infected CRFK Cells

To investigate which phase of infection sensitization with fADSC-CM is needed to inhibit FCV and FHV1 replication, cells were exposed to naïve fADSC-CM at different phases during virus infection. When the CRFK cells were exposed to fADSC-CM after viral infection, the FCV and FHV1 vmRNA levels were 43.8% and 56.7% compared with those of the mock CRFK cells (Figure 5). Moreover, when the CRFK cells were exposed to fADSC-CM both before and after virus infection, the FCV and FHV1 vmRNA levels showed a greater decrease, at 30.2% and 34.1%, respectively. However, when the CRFK cells were treated with fADSC-CM during virus attachment and entry, the FCV and FHV1 vmRNA levels increased (196.3% and 127.7%), but vmRNA levels decreased when the CRFK cells were exposed to fADSC-CM in all phases during the course of infection with both FCV and FHV1, at 41.5% and 43.9%, respectively.

The results of the plaque assay followed a similar trend to the expression of vmRNA levels in the CRFK cells (Figure 6). When virus-infected cells were sensitized to fADSC-CM after infection, the infectious virus titers in the supernatant, measured in plaque-forming units (PFU) for FCV (9.15 × 10^5^ PFU/mL) and FHV1 (1.05 × 10^6^ PFU/mL), were decreased compared with those of the mock CRFK cells (FCV: 2.22 × 10^6^ PFU/mL, FHV1: 2.19 × 10^6^ PFU/mL). Moreover, when fADSC-CM was used to sensitize the CRFK cells both before and after virus infection, the titers of FCV (8.20 × 10^5^ PFU/mL) and FHV1 (8.88 × 10^5^ PFU/mL) decreased further.

### 3.7. RNA Sequencing

RNA-seq identified global changes in gene expression of the CRFK cells sensitized with fAD-MSC-CM compared with gene expression in the mock CRFK cells (Figure 7). After comparing between the groups and sorting the data according to the above requirements, 181 upregulated genes and 229 downregulated genes were identified within a total of 13,501 detected genes. The top 15 genes showing the highest fold changes are listed in Table 2 and Table 3, and all upregulated and downregulated genes are shown in Appendix A.

### 3.8. Enrichment Analysis

Functional enrichment analysis was performed using 181 selected upregulated genes and 229 downregulated genes. Analysis of the upregulated genes identified one GO molecular function (MF), 44 GO biological process (BP), 7 GO cellular component (CC), and 1 KEGG pathway. The analysis of downregulated genes identified 6 GO MF, 150 GO BP, and 4 GO CC. The categories of all GO MF, GO CC, and KEGG pathways and the top 10 GO BP are shown in Figure 8 and Figure 9 and Table 4 and Table 5. All results, including annotated GO terms, are listed in Appendix A.

GSEA was performed using the result of RNA-seq. In total, 43 pathways were determined to be significantly altered in the CRFK cells sensitized with fADSC-CM compared with pathways in the mock CRFK cells (Table 6). Figure 10 shows a RUG plot of the top 10 pathway and network functions.

## 4. Discussion

MSCs and/or their secretomes have been investigated as methods of therapy used in several animal models of viral disease [9]. These studies have demonstrated that MSC treatment effectively suppresses inflammation and enhances immunity. Moreover, some studies have shown that microRNAs (miRNAs), including those in exosomes secreted by MSCs, directly inhibit viral expression and replication [28,29]. In the veterinary field, FCV and FHV1 are associated with the onset and severity of feline chronic gingivostomatitis and upper respiratory infections, and some studies have shown the usefulness of MSCs for therapy against these conditions [21,22,23]. These studies primarily discussed improving clinical symptoms dampening inflammation and did not examine the effects on the viruses themselves. Therefore, we investigated the effects of MSC secretome at different protein concentrations and molecular weights on FCV and FHV1 in this study.

Most studies have focused on the antiviral effects of miRNAs contained in exosomes secreted from MSCs [9,28,29,30], but the antiviral effects of MSC secretomes have not been investigated. Therefore, naïve feline AD-MSC secretome were collected as CM in this study. Based on the protein concentrations of the naïve fADSC-CM, the highest concentration used in the comparison experiments was 200 µg/mL. Several methods of purifying exosomes from CM have been published, and the use of MWCO ultrafiltration has been reported to increase exosome recovery [7,31,32]. In this study, vmRNA expression in the CRFK cells sensitized to naïve fADSC-CM showed the greatest inhibition of both FCV and FHV1 compared with the expression in the CRFK cells sensitized to 3 kDa, 30 kDa, and 100 kDa MWCO-concentrated fADSC-CM. We performed our experiments based on naïve fADSC-CM. Therefore, it is possible that the number of exosomes was not so different from that in naïve fADSC-CM. From the results of the dose-dependent suppression of viral replication, therefore, FCV and FHV1 may also be inhibited by exosomes secreted from fAD-MSCs. To clarify the antiviral effects of feline AD-MSCs exosomes on FCV and FHV1, further comparative experiments using purified feline AD-MSCs exosomes and CM excluded exosomes are needed.

The results of sensitizing cells to CM during different phases of the viral infection process showed that CM did not inhibit viral entry but did inhibit both FCV and FHV1 replication. These effects were similar to the antiviral effects of exosomal miRNAs demonstrated on hepatitis C virus, which showed inhibited viral replication after infection of the cells [28]. In both FCV and FHV1, vmRNA expressions were increased in the CRFK cells sensitized at the virus attachment and entry phases. Feline junctional adhesion molecule (JAM)-1, an immunoglobulin-like protein present in tight junctions, was identified as a cellular-binding molecule of the FCV F4 strain [33]. In our RNA-seq results, we detected no *JAM-1* expression in the CRFK cells, but there was no difference in *JAM-2* or *JAM-3* expression between the mock CRFK cells and the CRFK cells sensitized with fADSC-CM. In FHV1, envelope glycoproteins play important roles in cell attachment and entry, and heparin sulfate on the host cell surface serves as a receptor [34,35]. RNA-seq of the CRFK cells detected some heparin-sulfate-related genes, but there were no changes in the expression of these in the CRFK cells sensitized with fADSC-CM compared with the expression of these in mock CRFK cells. It is unclear why FCV and FHV1 entry into the CRFK cells was facilitated, but we speculated that factors other than host cell receptors may have been involved.

GSEA showed that fADSC-CM altered various functions of CRFK cells, such as metabolic pathways, signaling pathways, the cell cycle, the lysosome, oxidative phosphorylation, and the phagosome. Although RNA-seq did not determine the mechanisms involved in the inhibition of viral replication, the observation that fADSC-CM inhibited the viral replication of two different type of viruses, FCV, which is a single-positive-stranded RNA non-enveloped virus, and FHV1, which is a double-stranded DNA enveloped virus, was very interesting.

In conclusion, our findings revealed that fADSC-CM had antiviral effects by inhibiting FCV and FHV1 replication. It is suggested that AD-MSC therapy for feline chronic gingivostomatitis and upper respiratory infection may be effective due not only to the anti-inflammatory effects, as already reported, but also effects that prevent viral replication. Further studies are needed to clarify the mechanisms by which fADSC-CM prevents viral replication.

## Figures and Tables

**Figure 1 viruses-14-01687-f001:**
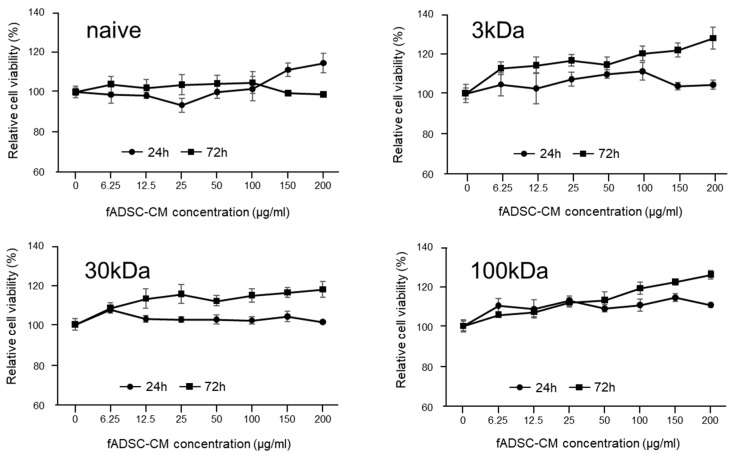
Cytotoxic effects of fADSC-CM treatment on CRFK cells. Cells were treated with 0–200 µg/mL of fADSC-CM for 24 h and 72 h. Relative cell viability was determined by CCK8 assay and normalized to the value of the 0 µg/mL group (set at 100%). Data are expressed as the mean ± standard deviation.

**Figure 2 viruses-14-01687-f002:**
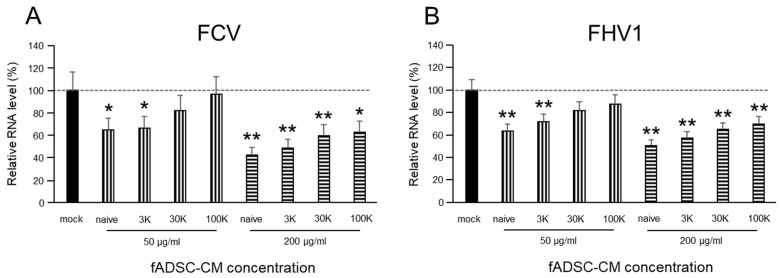
Evaluation of the effects of fAD-MSC-CM on viral replication (**A**) Relative FCV viral mRNA levels in CRFK cells. (**B**) Relative FHV1 viral mRNA levels in CRFK cells. Data are expressed as the mean ± standard deviation. Asterisks indicate significant differences compared with mock-treated group (*, *p* < 0.05; **, *p* < 0.01).

**Figure 3 viruses-14-01687-f003:**
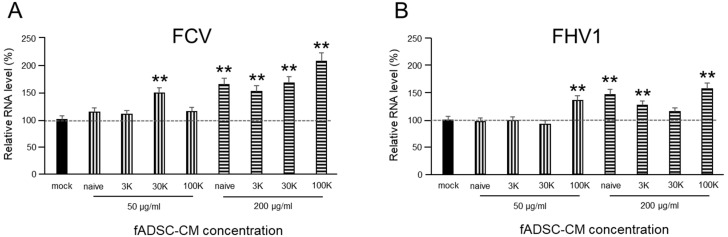
Evaluation of the effects of fAD-MSC-CM on viral entry (**A**) Relative FCV mRNA levels in CRFK cells. (**B**) Relative FHV1 mRNA levels in CRFK cells. Data are expressed as the mean ± standard deviation. Asterisks indicate significant differences compared with mock-treated group (**, *p* < 0.01).

**Figure 4 viruses-14-01687-f004:**
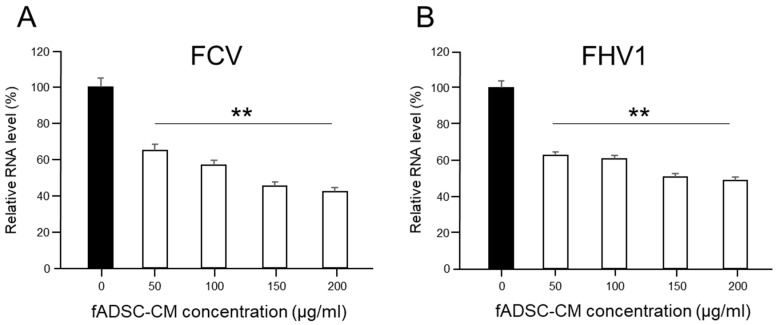
Dose-dependent effects of fADSC-CM on viral replication (**A**) Relative FCV viral mRNA levels in CRFK cells. (**B**) Relative FHV1 viral mRNA levels in CRFK cells. Data are expressed as the mean ± standard deviation. Asterisks indicate significant differences compared with mock-treated group (**, *p* < 0.01).

**Figure 5 viruses-14-01687-f005:**
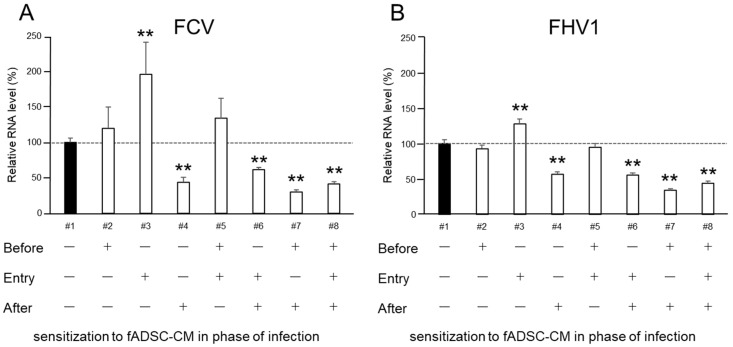
Effects of fADSC-CM on viral replication in CRFK cells cultured with (+) or without (−) naïve fADSC-CM before viral infection, during infection (attachment and entry), and/or after infection (**A**) Relative FCV viral mRNA levels in CRFK cells. (**B**) Relative FHV1 viral mRNA levels in CRFK cells. Data are expressed as the mean ± standard deviation. Asterisks indicate significant differences compared with mock-treated group (**, *p* < 0.01).

**Figure 6 viruses-14-01687-f006:**
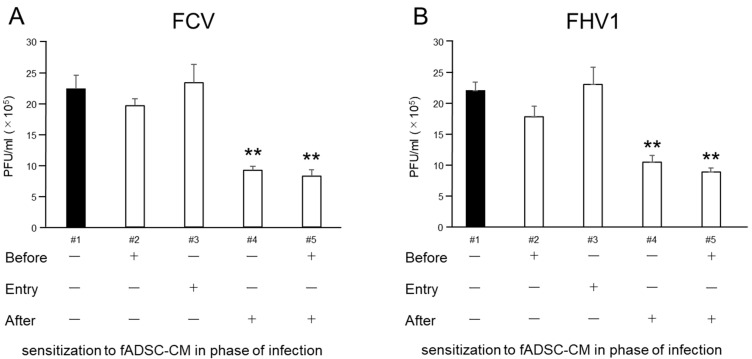
Comparison of virus titers measured in PFU in the supernatant of infected CRFK cells sensitized with (+) or without (-) fADSC-CM at different infection phases. Virus titers of FCV (**A**) and FHV1 (**B**). Data are expressed as the mean ± standard deviation. Asterisks indicate significant differences compared with mock-treated group (**, *p* < 0.01).

**Figure 7 viruses-14-01687-f007:**
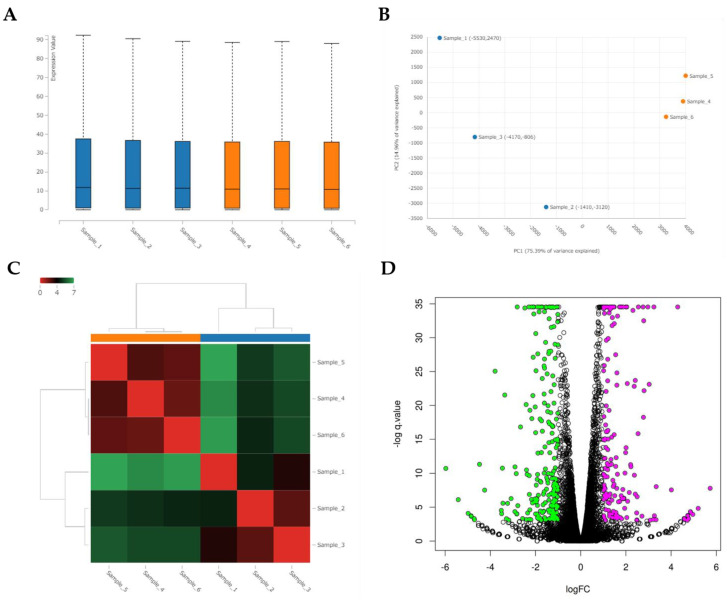
Analysis of RNA sequencing of CRFK cells sensitized with naïve fADSC-CM (**A**) Boxplot of the expression values for each sample. (**B**) PCA plot of samples. (**C**) Heatmap of expression similarity among samples. (**D**) Volcano plot of each group. Magenta dots indicate upregulated genes and green dots indicate downregulated genes in CRFK treated with fADSC-CM, according to adjusted *p*-value < 0.05 and fold change > 2 or <0.5. (**E**) Heatmap of expression values of selected genes in each sample (showing only the first 100 genes). Samples 1–3, mock-treated CRFK cells; samples 4–6, CRFK cells treated with fADSC-CM.

**Figure 8 viruses-14-01687-f008:**
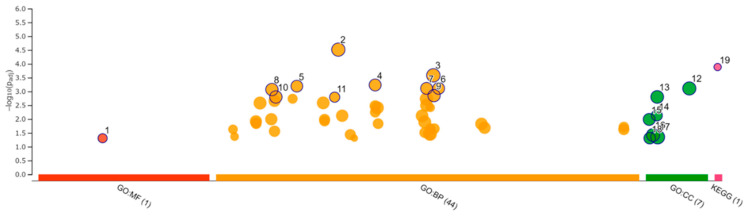
Functional enrichment analysis of 181 upregulated differentially expressed genes by g:Profiler. Results of enrichment analysis presented in the form of a Manhattan plot, where the *x*-axis shows the functional terms grouped by the color-coded of source database, while the *y*-axis shows the enrichment adjusted *p*-values in negative decimal logarithm scale. Dots in the graph indicate all enriched terms meeting the significance criterion of *p* < 0.05.

**Figure 9 viruses-14-01687-f009:**
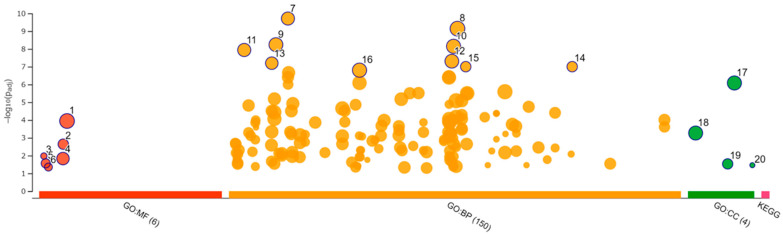
Functional enrichment analysis of 229 downregulated differentially expressed genes by g:Profiler. Results of enrichment analysis presented in the form of a Manhattan plot, where the *x*-axis shows the functional terms grouped by the color-coded source database, while the *y*-axis shows the enrichment adjusted *p*-values in negative decimal logarithm scale. Dots in the graph indicate all enriched terms meeting the significance criterion of *p* < 0.05.

**Figure 10 viruses-14-01687-f010:**
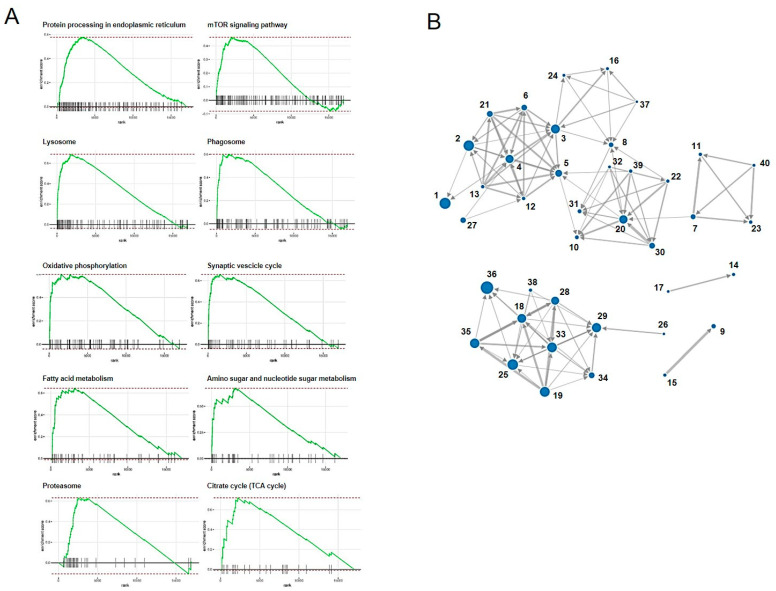
Top 10 RUG plots of the pathways (**A**) and network functions (**B**).

**Table 1 viruses-14-01687-t001:** Primer and probe sequences for real-time RT-PCR.

Target	Oligonucleotide Sequence	
FCV F4	Forward	5′-TCGATTCCTTCGGACCTGATC-3′
	Reverse	5′-AAGTCGAAATGACGGTTTGCTT-3′
	Probe	FAM-TAATCGCTACTGGACTGAC-TAMRA
FHV1 gC	Forward	5′-ACGGGAAGCCAATAGAAAG-3′
	Reverse	5′-CGGAATAGCCAACACAGAA-3′
	Probe	FAM-ATGAGTCCATCTATCCATACACTTGCCG-TAMRA
GUSB	Forward	5′-CTACATCGATGACATCACCATCAG-3′
	Reverse	5′-CGCCTTCAACAAAAATCTGGTAA-3′
	Probe	FAM-ACCAGCGTGAACCAAGACACTGGGC-TAMRA

**Table 2 viruses-14-01687-t002:** Top 15 most highly upregulated genes ranked by increasing fold change in expression.

Gene ID	Gene Symbol	Fold Change	Padj	Gene Function
ENSFCAG00000035883	PKLR	53.00	0.0004143	pyruvate kinase L/R
ENSFCAG00000010889	ADAMTS20	36.72	0.0079952	ADAM metallopeptidase with thrombospondin type 1 motif 20
ENSFCAG00000034557	PDX1	31.17	0.0195328	pancreatic and duodenal homeobox 1
ENSFCAG00000031865	CFAP97D1	29.18	0.0144898	CFAP97 domain containing 1
ENSFCAG00000031341	TMPRSS11E	28.93	0.0130241	transmembrane serine protease 11E
ENSFCAG00000004768	SHANK2	27.41	0.0366154	SH3 and multiple ankyrin repeat domains 2
ENSFCAG00000037311	RNase_MRP	26.70	0.0240336	RNase MRP
ENSFCAG00000024035	PAQR9	26.63	0.0228903	progestin and adipoQ receptor family member 9
ENSFCAG00000043275	CAMSAP3	26.29	0.0470668	calmodulin regulated spectrin associated protein family member 3
ENSFCAG00000041431	CXCR5	24.09	0.0370509	C-X-C motif chemokine receptor 5
ENSFCAG00000011344	SLC13A3	24.09	0.0410041	solute carrier family 13 member 3
ENSFCAG00000015185	BPIFB4	24.04	0.0369054	BPI fold containing family B member 4
ENSFCAG00000045063	HES3	23.82	0.0356022	hes family bHLH transcription factor 3
ENSFCAG00000041480	SLAMF7	19.60	2.676 × 10^−27^	SLAM family member 7
ENSFCAG00000010157	NCKAP1L	16.13	0.0005246	NCK associated protein 1 like

**Table 3 viruses-14-01687-t003:** Top 15 most highly downregulated genes ranked by decreasing fold change in expression.

Gene ID	Gene Symbol	Fold Change	Padj	Gene Function
ENSFCAG00000045199	ARX	0.016	2.198 × 10^−5^	aristaless related homeobox
ENSFCAG00000013814	SPTSSB	0.023	0.002199	serine palmitoyltransferase small subunit B
ENSFCAG00000043019	TNMD	0.032	0.0167252	tenomodulin
ENSFCAG00000008166	S100A14	0.035	0.024189	S100 calcium binding protein A14
ENSFCAG00000001649	CDHR1	0.035	0.0257187	cadherin related family member 1
ENSFCAG00000004983	HTR3A	0.038	0.0357949	5-hydroxytryptamine receptor 3A
ENSFCAG00000028760	CB2H6orf52	0.038	0.0429303	chromosome B2 C6orf52 homolog
ENSFCAG00000000794	CCDC85A	0.045	1.199 × 10^−5^	coiled-coil domain containing 85A
ENSFCAG00000004322	TENM4	0.052	0.0005413	teneurin transmembrane protein 4
ENSFCAG00000022100	PLEKHS1	0.072	1.323 × 10^−11^	pleckstrin homology domain containing S1
ENSFCAG00000033516	ZC2HC1B	0.087	0.0107755	zinc finger C2HC-type containing 1B
ENSFCAG00000029875	RERG	0.088	0.0177269	RAS like estrogen regulated growth inhibitor
ENSFCAG00000027949	ART4	0.090	2.117 × 10^−5^	ADP-ribosyltransferase 4
ENSFCAG00000004490	PLEKHB1	0.092	0.0037363	pleckstrin homology domain containing B1
ENSFCAG00000045651	ANGPT1	0.097	4.458 × 10^−10^	angiopoietin 1

**Table 4 viruses-14-01687-t004:** Top 10 of each gene ontology (GO) molecular function (MF), biological processes (BP), and cellular component (CC) and KEGG terms.

ID	Category	GO ID	Term	Padj
1	GO:MF	GO:0019955	cytokine binding	4.93 × 10^−2^
2	GO:BP	GO:0032502	developmental process	3.06 × 10^−5^
3	GO:BP	GO:0051179	localization	2.61 × 10^−4^
4	GO:BP	GO:0040011	locomotion	5.82 × 10^−4^
5	GO:BP	GO:0016477	cell migration	6.42 × 10^−4^
6	GO:BP	GO:0051674	localization of cell	7.73 × 10^−4^
7	GO:BP	GO:0048870	cell motility	7.73 × 10^−4^
8	GO:BP	GO:0009653	anatomical structure morphogenesis	8.51 × 10^−4^
9	GO:BP	GO:0051239	regulation of multicellular organismal process	1.41 × 10^−3^
10	GO:BP	GO:0010033	response to organic substance	1.58 × 10^−3^
11	GO:BP	GO:0032103	positive regulation of response to external stimulus	1.61 × 10^−3^
12	GO:CC	GO:0071944	cell periphery	7.78 × 10^−4^
13	GO:CC	GO:0012505	endomembrane system	1.58 × 10^−3^
14	GO:CC	GO:0009986	cell surface	7.39 × 10^−3^
15	GO:CC	GO:0005576	extracellular region	1.03 × 10^−2^
16	GO:CC	GO:0005886	plasma membrane	3.77 × 10^−2^
17	GO:CC	GO:0016020	membrane	4.46 × 10^−2^
18	GO:CC	GO:0005615	extracellular space	4.91 × 10^−2^
19	KEGG	KEGG:03320	PPAR signaling pathway	1.30 × 10^−4^

**Table 5 viruses-14-01687-t005:** Top 10 of each gene ontology (GO) molecular function (MF), biological processes (BP), and cellular component (CC) terms.

ID	Category	GO ID	Term	Padj
1	GO:MF	GO:0005515	protein binding	1.13 × 10^−4^
2	GO:MF	GO:0005126	cytokine receptor binding	2.27 × 10^−3^
3	GO:MF	GO:0001730	2′−5′-oligoadenylate synthetase activity	1.06 × 10^−2^
4	GO:MF	GO:0005102	signaling receptor binding	1.47 × 10^−2^
5	GO:MF	GO:0003725	double-stranded RNA binding	2.72 × 10^−2^
6	GO:MF	GO:0003950	NAD + ADP-ribosyltransferase activity	4.41 × 10^−2^
7	GO:BP	GO:0009605	response to external stimulus	1.95 × 10^−10^
8	GO:BP	GO:0050896	response to stimulus	7.23 × 10^−10^
9	GO:BP	GO:0007275	multicellular organism development	5.75 × 10^−9^
10	GO:BP	GO:0048856	anatomical structure development	7.12 × 10^−9^
11	GO:BP	GO:0002376	immune system process	1.15 × 10^−8^
12	GO:BP	GO:0048731	system development	4.90 × 10^−8^
13	GO:BP	GO:0006955	immune response	6.33 × 10^−8^
14	GO:BP	GO:0140546	defense response to symbiont	9.95 × 10^−8^
15	GO:BP	GO:0051607	defense response to virus	9.95 × 10^−8^
16	GO:BP	GO:0032502	developmental process	1.57 × 10^−7^
17	GO:CC	GO:0071944	cell periphery	8.23 × 10^−7^
18	GO:CC	GO:0005886	plasma membrane	5.39 × 10^−4^
19	GO:CC	GO:0062023	collagen-containing extracellular matrix	2.93 × 10^−2^
20	GO:CC	GO:1990584	cardiac Troponin complex	3.45 × 10^−2^

**Table 6 viruses-14-01687-t006:** Significantly enriched gene sets in CRFK cells sensitized with fADSC-CM compared with those in mock-treated CRFK cells using RaNa-seq.

No	Pathway_ID	Pathway	NumGenes	ES	NES	Size	Padj
1	fca04141	Protein processing in endoplasmic reticulum	67	0.577	2.182	142	0.005
2	fca04150	mTOR signaling pathway	40	0.464	1.736	132	0.005
3	fca04142	Lysosome	52	0.688	2.498	108	0.005
4	fca04145	Phagosome	31	0.601	2.143	94	0.005
5	fca00190	Oxidative phosphorylation	25	0.597	2.053	75	0.005
6	fca04721	Synaptic vesicle cycle	20	0.66	2.148	55	0.005
7	fca01212	Fatty acid metabolism	23	0.645	2.023	45	0.005
8	fca00520	Amino sugar and nucleotide sugar metabolism	21	0.674	2.061	39	0.005
9	fca03050	Proteasome	28	0.631	1.894	35	0.005
10	fca00020	Citrate cycle (TCA cycle)	13	0.719	2.038	27	0.005
11	fca01040	Biosynthesis of unsaturated fatty acids	13	0.726	1.926	20	0.005
12	fca04966	Collecting duct acid secretion	10	0.764	2.048	21	0.005
13	fca_M00160	V-type ATPase, eukaryotes	12	0.815	2.185	21	0.005
14	fca00100	Steroid biosynthesis	11	0.866	2.185	16	0.005
15	fca_M00341	Proteasome, 19S regulatory particle (PA700)	17	0.782	1.999	17	0.005
16	fca00531	Glycosaminoglycan degradation	10	0.83	2.032	14	0.005
17	fca_M00101	Cholesterol biosynthesis, squalene 2,3-epoxide => cholesterol	8	0.965	2.096	9	0.005
18	fca05160	Hepatitis C	25	−0.465	−1.814	107	0.007
19	fca05168	Herpes simplex infection	30	−0.443	−1.766	122	0.007
20	fca01200	Carbon metabolism	35	0.496	1.783	99	0.007
21	fca05323	Rheumatoid arthritis	17	0.571	1.885	59	0.007
22	fca_M00001	Glycolysis (Embden-Meyerhof pathway), glucose => pyruvate	11	0.733	1.921	19	0.008
23	fca00062	Fatty acid elongation	12	0.697	1.868	21	0.01
24	fca00604	Glycosphingolipid biosynthesis—ganglio series	7	0.786	1.887	13	0.01
25	fca04390	Hippo signaling pathway	52	−0.406	−1.642	134	0.01
26	fca_M00285	MCM complex	6	−0.919	−1.838	6	0.011
27	fca04976	Bile secretion	14	0.55	1.779	53	0.011
28	fca05162	Measles	20	−0.454	−1.733	94	0.013
29	fca04110	Cell cycle	45	−0.434	−1.7	112	0.013
30	fca01230	Biosynthesis of amino acids	23	0.552	1.808	57	0.013
31	fca00051	Fructose and mannose metabolism	15	0.636	1.826	29	0.013
32	fca_M00003	Gluconeogenesis, oxaloacetate => fructose-6P	7	0.797	1.876	12	0.016
33	fca05161	Hepatitis B	23	−0.405	−1.608	121	0.019
34	fca04115	p53 signaling pathway	19	−0.506	−1.777	58	0.023
35	fca05164	Influenza A	24	−0.412	−1.632	116	0.024
36	fca04015	Rap1 signaling pathway	27	−0.364	−1.519	165	0.024
37	fca_M00079	Keratan sulfate degradation	5	0.92	1.717	5	0.024
38	fca_M00679	BMP signaling	8	−0.707	−1.884	17	0.024
39	fca_M00009	Citrate cycle (TCA cycle, Krebs cycle)	11	0.668	1.789	21	0.024
40	fca_M00415	Fatty acid biosynthesis, elongation, endoplasmic reticulum	10	0.757	1.82	13	0.03
41	fca_M00077	Chondroitin sulfate degradation	4	0.909	1.698	5	0.032
42	fca04152	AMPK signaling pathway	31	0.435	1.586	110	0.035
43	fca04964	Proximal tubule bicarbonate reclamation	5	0.725	1.775	14	0.04

## Data Availability

All the data are shown in the manuscript and can be obtained from the authors on request.

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
