# Peer review of "Antiviral Effects of Adipose Tissue-Derived Mesenchymal Stem Cells Secretome against Feline Calicivirus and Feline Herpesvirus Type 1"

_viruses, 2022, doi:10.3390/v14081687_

Round 1

Reviewer 1 Report

The manuscript is well written and well structured and in my opinion merits publication after some minor rectifications. 

As per the title and the abstract the study main focus was evaluating the effect of AD-MSC CM on FCV and FHV1. As well as this, the authors have also provided extensive amount of data associated with this study and that made me go back to check what was the aim of the study. This included cytotoxicity, enrichment, gene expression data, up-regulation and down-regulation data ...etc seen in figures 7, 8, 9 and 10 as well as tables 2-6. I think this adds weight to the study but perhaps this might need to be added at the beginning too as an aim of the study.

There are some minor grammatical structural and spelling issues that should be addressed throughout the manuscript. 

Review also the reference list. There are a couple of capitalisations needed. 

I have attached my comments in the PDF manuscript version.

Author Response

We are pleased to note the favorable comments of reviewers in attached file and have made corrections that we hope meet with your approval. Revisions in the manuscript are shown in yellow. We hope that you find the revised manuscript acceptable for publication in Viruses. We look forward to hearing from you at your earliest convenience.

Reviewer 2 Report

Dear editor, I reviewed the manuscript entitled: “ Antiviral effects of adipose tissue-derived mesenchymal stem cells  secretomes against feline calicivirus and feline herpesvirus type 1, with great interest. The paper is well written and very timely. The work that was done appear solid. I have some minor input and a question. Otherwise, this paper will be an asset to the field of adMSCs and particularly, to the field of viral infections and MSCs interactions.

1.    Abstract and elsewhere: To date, FCV was not found to cause’ feline gingivostomatitis but only ‘associated’ with the disease. (although I do agree with the sentiment of the authors that this may be the case).

Fried, W.A., et al., Use of unbiased metagenomic and transcriptomic analyses to investigate the association between feline calicivirus and feline chronic gingivostomatitis in domestic cats. Am J Vet Res, 2021. 82(5): p. 381-394.

2.    During adMSCs culture, did the authors noticed any deleterious effect of feline foamy virus? Especially in P3?

Arzi, B., et al., Feline foamy virus adversely affects feline mesenchymal stem cell culture and expansion: implications for animal model development. Stem Cells Dev, 2015. 24(7): p. 814-23.

Author Response

(The authors gave the same response as above.)
